# A Taxonomy of Crisis Management Functions

**Todor Tagarev *** 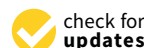 **and Valeri Ratchev**

Institute of Information and Communication Technologies, Bulgarian Academy of Sciences, 1113 Sofia, Bulgaria; ratchevv@gmail.com
* Correspondence: tagarev@bas.bg

**Abstract:** The management of crises triggered by natural or manmade events requires a concerted effort of various actors crossing institutional and geographic boundaries. Technological advances allow to make crisis management more effective, but innovation is hindered by dispersed and often disconnected knowledge on the lessons learned, gaps, and solutions. Taxonomies enable the search for information of potential interest. This article presents a taxonomy of crisis management functions, designed on the basis of a conceptual model integrating the concepts of hazard, vulnerability, risk, and community, and the main consequence- and management-based concepts. At its highest level, the taxonomy includes ten functional areas: preparatory (mitigation, capability development, and strategic adaptiveness), operational (protection, response, and recovery), and common (crisis communications and information management; command, control, and coordination; logistics; and security management). The taxonomy facilitates the navigation of online platforms and the matching of needs and solutions. It has broader applications, e.g., for structuring the assessment of the societal impact of crisis management solutions and as a framework for a comprehensive assessment of disaster risk reduction measures. While the taxonomy was developed within a research and innovation project supported by the European Union, it reflects and is compatible with established international concepts and classification schemes, and is thus applicable by a wider international community.

**Keywords:** crisis management; emergency management; disaster risk reduction; climate change adaptation; resilience; taxonomy

## 1. Introduction

Interrelated consequences of human activities have led to a steady rise in the exposure to and impact of natural and manmade disasters [1]. Aware of the effects of intensive industrialisation, uncontrolled urbanisation, ecosystem degradation, and climate change, the international community undertook initiatives to help states prevent and mitigate risks, develop and implement appropriate preparedness and effective response to disasters, and thus to achieve sustainable development goals [2]. The Sendai Framework 2015–2030 [1] provided the ground for extensive and systematic work on disaster risk reduction and strengthening societal resilience. The European Union (EU) defined its priorities accordingly and implemented a risk-informed approach to policy making. Along with disaster risk reduction and resilience measures, the European Commission declared the development of comprehensive disaster preparedness for effective response and recovery, i.e., crisis management, capabilities as one of the principal priorities of the European Union [3].

The study of crisis management is becoming increasingly broad in scope. While the traditional focus on crises triggered by natural disasters and industrial catastrophes, and the capacity to respond and recover remains valid, policymakers and researchers are becoming more concerned with crises triggered by terrorist acts [4], cyberattacks [5], interdependencies among critical infrastructures and

cascading effects [6], and look into a more comprehensive approach to disaster risk reduction [7–9] and the need to adapt to climate change [10–12].

First responders and policymakers, assisted by researchers, identify new needs and requirements by studying the experience in crisis management, predicting hazards, or analysing vulnerabilities. In parallel, research institutes and commercial companies develop new or adapt existing crisis management solutions. Occasionally, the degree to which such solutions meet newly identified needs is established in a concept development and experimentation process [13] or in trials [14].

The body of knowledge on crisis management needs and potential solutions is, however, dispersed, and as a rule, disconnected. Therefore, the DRIVER+ project (Driving Innovation in Crisis Management for European Resilience) aimed, inter alia, to speed up market uptake and facilitate innovation by filling this gap and allowing practitioners to quickly find adequate solutions to their needs. Towards that aim, DRIVER+ developed methodological, technical and information infrastructure, including a trial guidance methodology, a pan-European testbed infrastructure (physical, methodological and technical infrastructure elements allowing to conduct trials and evaluate crisis management solutions within an appropriate environment in a systematic way [15]), and an online platform known as "Portfolio of Solutions" (POS).

The taxonomy of crisis management functions (the "CM taxonomy" for short), designed by the authors, is at the heart of the POS platform. It is designated to support the navigation of the platform and allow crisis management practitioners to find information on solutions that are most relevant to their particular needs. Furthermore, and in the pursuit of the long-term use of the taxonomy, we considered in its design some broader applications by:

- Providing a platform for sharing crisis management related knowledge in terms of experience, solutions on practical problems, development of instruments such as doctrines, procedures, and equipment;
- Facilitating professional communications and information sharing among various crisis management stakeholders, between and within specialised organisations, trainers, research communities, industry, software producers, and other actors, as well as throughout the European community and citizens for strengthening disaster response volunteerism and resilience;
- Affording the integration of various datasets;
- Providing decision support in every functional area of crisis management, as well as for most specific tasks;
- Offering semantic frameworks for the crisis management organisation of professional and volunteer formations, as well as for national crisis command and management architectures;
- Exploring the hazards-related professional vocabulary;
- Helping focused gap analyses throughout the spectrum of crisis management;
- Representing case-specific semantics for the development of solutions and tools;
- Providing a tool for the study of issues in disaster sociology and crisis management.

The CM taxonomy uses five classes of hazards and six properties of the concept of 'community' [16]. It has a hierarchical structure with ten functional areas, 54 functions, 262 sub-functions, and 103 tasks in its current version (ver. 2.1, May 2020). The decomposition is limited to the taxonomy's fourth level to make it manageable. However, if necessary, it could be further elaborated both horizontally and in-depth, down to tasks, activities, and transactions.

Section 2 of this article describes the methodological approach to the design of the taxonomy and includes references to relevant organisational and academic literature. Section 3 outlines the taxonomy, with an illustrative visual representation of the functional areas, and provides an analysis of compatibility with existing classification schemes. Section 4 discusses the taxonomy's current and future use and its governance. The current version of the taxonomy is included as a supplement to this article.

## 2. Design Methodology and Underlying Concepts

The taxonomy is built on the system theory as a mechanism for structuring the knowledge about the crisis management domain. The process of taxonomy development consists of information collection, systematic analysis, and classification of system attributes [17,18] and is presented in Figure 1.

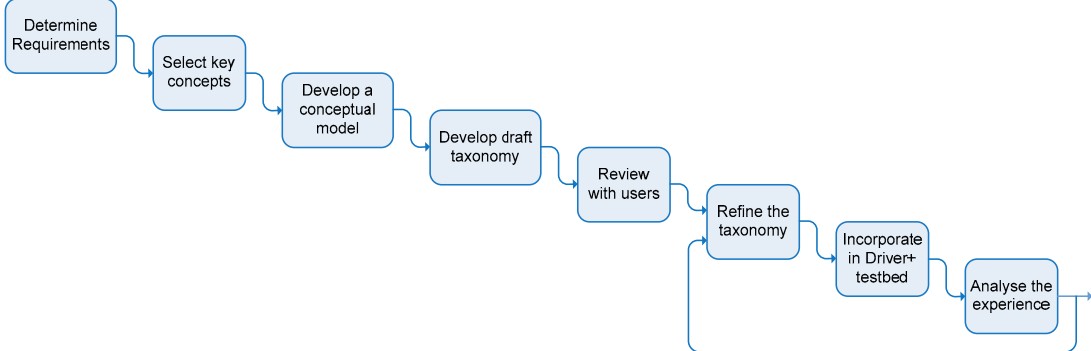

**Figure 1.** The process of developing the taxonomy of crisis management functions.

The selected method of work includes several tasks in the development of the taxonomy, presented in Figure 1, and maintaining it in its entire lifecycle. The method is adapted from several sources [17,19–22]. The detailed explanation starts with the requirements (Section 2.1), followed by the key underlying concepts (Section 2.1), and the conceptual model (Section 2.3) on which the taxonomy was built. The iterative design and review of the taxonomy are described in Section 4.

### 2.1. Requirements

In its first version, the taxonomy organised crisis management functions within the scope of crises caused by natural and manmade hazards, including the consequences of terrorist attacks (but not the anti-terrorism itself). Crisis management for civil protection is a complex endeavour, performed by multiple organisations, in a complex social environment, and with significant political meaning. Protecting lives and property, critical infrastructure, and the environment are functions of the highest importance and value. Following the International Organization for Standardization (ISO) standard ISO 15489-1 [23], the higher the risk level and accountability and/or public scrutiny is, the greater is the need for accuracy, precision, and control in preparing and organising crisis management operations and measures. To fulfil such requirements using the traditional classification of organisational structures and subjects is challenging, if at all possible.

On the one hand, the organisations engaged in the contemporary European management of crises are rather different and specifically organised, function under different legislation, apply different operational procedures, and use equipment that is rarely interoperable. An organisation-based classification would probably lead to more confusion than consolidation. On the other hand, classification under subject content fits more the classification of items; instead, crisis management is most of all about processes.

The functional taxonomy overcomes these concerns. Necessities determine functions, while different organisational arrangements may allow the implementation of the same function. For example, during the Cold War, the Bulgarian Civil Defence was a militarised structure within the Ministry of Defence implementing military operational procedures. Later, it became a civil service under the Ministry of Interior, with its own legal act. Nevertheless, its functions in support of the population have hardly been changed.

The functions provide information about the actors, character, direction, resources, and scope of processes. Their classification derives from the context of the process under consideration, e.g., the function 'planning' has a different context in the functional areas 'mitigation' and 'response.'

Accounting for context is very important for the overall security sector and policymaking. National systems and policies for 'national security,' 'national defence,' 'national crisis management,' etc., are interconnected and functionally bounded. Regarding the taxonomy, this requires maximum terminological interoperability, along with the other aspects of interoperability, such as doctrinal, technical, procedural and legal. For example, the militaries use three levels of military art—strategy, operational art, and tactics—depending on the level of engagement. If the levels are changed to 'strategy, tactics, and operations' [24], then many processes like 'intelligence' and 'planning,' 'transportation' and 'communication' would have different content.

The taxonomy should be sufficiently detailed to allow the effective matching of needs and solutions while capturing high-priority functions, critical for the crisis management mission. The depth of their decomposition should not make the taxonomy hard to use, and the key processes need to be apparent to both practitioners and solution providers.

The taxonomy contributes to making future CM solutions available, accessible, usable, and re-usable. It has two main groups of users: solution providers who are mostly engineers, software developers, business entrepreneurs, researchers, etc., and policymakers, public administrators, executive officers, fire-fighters, medical doctors. The vocabularies of the two groups differ. While aiming to make the taxonomy definitions easy to understand by the two groups, this is not always possible, and therefore priority is given to the demand side—the crisis management practitioners who may use the POS platform even in critical moments.

## 2.2. Key Concepts

The initial scope of the taxonomy covered natural and manmade hazards and related disasters, and communities, critical infrastructure and assets, as well as the environment. The 'space' between hazards and communities is the crisis management functional domain—a functional area to which crisis management authorities, organisations, communities, and individuals have to allocate their activities to mitigate hazards and enhance public resilience, to build relevant capabilities and provide civil protection, and to establish control in cases of disasters and crises.

The key concepts shall encompass the crisis management domain. The term 'concept' is preferred instead of 'definition' (as a statement of the meaning), as it explains the relationships, e.g., between information and awareness; awareness and planning; planning and decisions; decisions and organisations; organisations and activities; activities and outcomes. The relationships serve as arguments for defining the taxonomy's scope and for building its hierarchy (the 'taxonomy tree'). Specific relationships emerge since 'disaster' is not merely a 'big incident,' and 'catastrophe' is not just a bigger disaster [25].

The conceptual approach in the crisis management domain involves turning abstract concepts into a framework, within which tools and solutions are and will be applied to collect and process information, take decisions, apply measures and perform operations. This framework is used to solve the fundamental problem of crisis management—to protect communities and their environment from hazards.

The following general framework was used by the authors to build the conceptual model of the CM taxonomy:

> Communities of people with their properties (public and private, cultural, infrastructure and assets, resources); state and private livelihoods; the work of commercial, administrative and non-for-profit organisations; governmental, commercial, and voluntary services; and environment (natural and built) may be considered as a system. The socio-political context influences this system, which may generate and is exposed to hazards and threats. A set of socio-political instruments must be built and used to minimise the risks from identified hazards and threats. In case risk management fails and communities enter a crisis, then measures to respond effectively, provide relief and recovery should be available and promptly undertaken. The understanding of causal relationships in public

protection facilitates the decision making on hazards' mitigation, reduction of vulnerabilities, strengthening response and recovery capabilities and readiness, and generally, in building community resilience.

The following key concepts are briefly explained within this conceptual framework.

2.2.1. Hazard, Vulnerability, and Risk

For crisis management in the civil protection context, 'hazard,' 'vulnerability,' and 'risk' are interrelated concepts. This triad determines strategy and policy, priorities, the allocation of resources, and the organisations needed for adequate preparation, protection, response, and recovery against and from hazards and threats, i.e., for crisis management. However, neither are 'hazard' and 'risk' synonyms nor can they be estimated without assessing the 'vulnerability' of every asset against a concrete hazard.

'Hazard' is defined by ISO 22300 as a 'source of potential harm' [26]. Within the aforementioned conceptual framework, a hazard has the potential to cause harm to people, their activities, property, and environment. A hazard is a "dangerous phenomenon, substance, human activity or condition that may cause loss of life, injury or other health impacts, property damage, loss of livelihoods and services, social and economic disruption, or environmental damage" [27].

Most of the official and research qualifications of hazards are based on their origin: natural; manmade, civil or technological; environmental; biological; and other variations [28]. A review by the European Commission of national risk assessments of the member states identified 25 natural and manmade hazards, among which the following 12 are addressed most often:

- "Natural hazards: Floods; Severe weather; Wild/Forest fires; Earthquakes; Pandemics/epidemics; and Livestock epidemics.
- Manmade hazards: Industrial accidents; Nuclear/radiological accidents; Transport accidents; Loss of critical infrastructure; Cyberattacks; and Terrorist attacks" [29].

Recent research has evolved in three directions: expanding the scope of classification through the further specialisation of hazards as, e.g., medical, transport, energy, limiting the classification list to natural and manmade hazards, despite that natural events are hazards only if they affect communities of people and their environment [30]; and looking for a different classification approach to reflect the complex origin of contemporary hazards [31,32].

For the purpose of designing a taxonomy of functions, the classical approach is reasonable and valuable, as it facilitates the consideration of cause–consequence relationships, helps to determine concrete case-specific mitigation and response measures for specific hazards, provides better hazard awareness to wider stakeholders, and last but not least, serves better the purpose of classifying crisis management solutions in a manner that is convenient for practitioners. The conceptual model reflects the following five classes of hazards:

- A natural hazard is an "unexpected and/or uncontrollable natural event of an unusual magnitude that might threaten people" [33];
- A technological hazard is "a hazard originating from technological or industrial conditions, including accidents, dangerous procedures, infrastructure failure or specific human activities, that may cause loss of life, injury, illness or other health impacts, property damage, loss of livelihoods and services, social and economic disruption, or environmental damage" [27];
- A biological hazard is a "process or phenomenon of organic origin or conveyed by biological vectors, including exposure to pathogenic micro-organisms, toxins and bioactive substances that may cause loss of life, injury, illness or other health impacts, property damage, loss of livelihoods and services, social and economic disruption, or environmental damage" [27];
- An environmental hazard is "any single or combination of toxic chemical, biological, or physical agents in the environment, resulting from human activities or natural processes, that may impact

the health of exposed subjects, including pollutants such as heavy metals, pesticides, biological contaminants, toxic waste, industrial and home chemicals" [27];

- A socio-natural hazard is "the phenomenon of increased occurrence of certain geophysical and hydro-meteorological hazard events, such as landslides, flooding, land subsidence and drought, that arise from the interaction of natural hazards with overexploited or degraded land and environmental resources" [27].

'Vulnerability' is a "physical feature or operational attribute that renders an entity open to exploitation or susceptible to a given hazard" [34], "the characteristics and circumstances of a community, system or asset that make it susceptible to the damaging effects of a hazard" [27]. It might result from ineffective governance, poor management, poor design, confusing realisation, or the badly performed operation of a community, organisation, asset, infrastructure, system, or network that can be disrupted by any hazard. Within the above established framework, "it is the interaction of the hazard of place (risk and mitigation) with the social profile of communities" [35].

In his review, Weichselgartner [36] identifies three perspectives on the community vulnerability to hazards. In the first perspective, vulnerabilities result from exposing the community's attributes to hazards; i.e., the hazards, with their characteristics, are the driver of vulnerability. The second perspective is on the community's capacity (organisation, resources, culture, etc.) to prepare for and resist hazards' repercussions. In the third perspective, the vulnerability of a concrete community or asset derives from its geographical location that presupposes the exposure to the dominant hazard(s). This triple perspective provides orientation about the conduct of risk assessment and risk management as components of the preparatory, operational, and recovery activities in crisis management.

'Risk' is a concept that, in the framework for taxonomy design, serves as a hub between hazard (threat) and vulnerability (community, assets, environment) assessment, on the one hand, and the comprehensive crisis management approach, on the other. As "risk is a combination of the consequences of an event (hazard) and the associated likelihood/probability of its occurrence" [37], the level of risk is a function of the likelihood (probability) of occurrence of a concrete hazard and the adverse impact (consequences) it may have to concrete objects—people, activities, assets, infrastructure, networks, resources, environment, etc. [38]. The *Risk Assessment and Mapping Guidelines for Disaster Management* of the European Commission elaborate further:

- "The terms 'probability' or 'likelihood' are understood as the probability or likelihood of the risk occurring or taking place in the future;
- 'Consequence' or 'impact' are understood as negative effects of the disaster or risk expressed in terms of human impacts, economic/infrastructure impacts and environmental impacts" [29].

According to ISO 31010, "risk assessment is the overall process of risk identification, risk analysis, and risk evaluation" [37]. Consequences may include "dead and missing; injury (mental and physical); disease (mental and physical); secondary hazards (fire, disease, etc.); contamination; displacement; breakdown in security; damage to infrastructure; breakdown in essential services; loss of property; loss of income . . . " [39]. Consequently, the logic of risk management (as a component of crisis management) is to reduce the likelihood of occurrence of concrete hazard (where possible) and to limit its consequences through eliminating vulnerabilities and strengthening the community's capabilities to resist, respond, and recover.

For the conceptual model of the functional taxonomy, two important conclusions derive from the 'hazard–vulnerability–risk' perspective. First, the information, analysis, and evaluation of hazards, vulnerabilities, and risks are the key factors of crisis management decision making and should be secured to the greatest extent in all normal, emergency, and critical conditions. Second, measures to put the hazards, vulnerabilities, and risks under maximum possible control are or should be undertaken throughout all preparatory and operational functional areas, with the overall aim to strengthen the community resilience.

### 2.2.2. Community

The concept of 'community' is central to the design of the taxonomy of crisis management functions. To reduce disaster risks, the role of the professional emergency management organisations is complemented by investments in building community resilience, trust, and social cohesion. From the taxonomy point of view, the concept of 'community' provides the necessary details on what concretely the crisis management functions (and respectively—tools and solutions) should focus on to maintain the people-centred prioritisation across preparatory and operational efforts.

According to the U.S. Department of Homeland Security (DHS) National Response Framework, communities are groups that share goals, values, and institutions. They are not always bound by geographic boundaries or political divisions. Instead, they may be faith-based organizations, neighborhood partnerships, advocacy groups, academia, social and community groups, and associations. Communities bring people together in different ways for different reasons, but each provides opportunities for sharing information and promoting collective action [40].

In this explanation, distinguishing the 'community' from the geographical area on which some people live is very important for the crisis management planners and leaders to prevent thinking that if there are people living in a particular area, then there is a community. There is no all-embracing 'community.' Rather, communities are defined by culture, interests, specific features (affection, interests, competition, status), as well as by a location that might be chosen as a result of the former determinants [41].

For the taxonomy model, the following key attributes of the concept of 'community' (adapted from several sources by the World Health Organization) are taken into consideration:

- The people in a concrete geographical area as demographics, respect to the safety regulations, willingness to participate in governmental mitigation and resilience-building programmes, attitude towards voluntarism and mutual help, historical experience, and emergency-related skills;
- Peoples' property, e.g., possession of infrastructure, land, forests, small dams, and the like, mobile communications and other assets usable for crisis management, public and cultural infrastructure within the area;
- Existing voluntary, formal, or not-for-profit organisations that might be involved in crisis management;
- Services provided by different jurisdictions, as well as commercial and voluntary services;
- The dominant sources of livelihood and their dependencies on plausible hazards;
- The living environment, both built and natural.

Reviewed sources are in agreement that there is no exemplary implementation of the culture of crisis response volunteerism and resilience in terms of better awareness, preparedness, and self-reliance in emergencies and crises. The role of the crisis management planners and authorities is to build an environment for and to support the establishment of such a culture. Towards that aim, specific tools and solutions should be developed or adapted to concrete types of communities.

### 2.2.3. Consequence-Based Concepts: Incident, Disaster, Catastrophe, and Crisis

The conceptual model underlying the CM taxonomy should provide opportunities for the classification of tools and solutions when hazards threaten or affect communities and their properties. Presumably, this limitation should put aside the 'incident' as it is a non-community situation. However, as history clearly illustrates, many community-wide and even international crises have begun as incidents, like those with chemical and nuclear facilities (e.g., the chemical industrial catastrophes in Seveso in 1975 and Bhopal, India in 1984, the Chernobyl nuclear power station catastrophe in 1986) that made people realise that European-wide policy on environmental protection was needed. Terminological clarity is essential for the taxonomy, as the conceptual model is comprehensive—in scope and in terms of levels of command and management—and needs a well established scale for

classification. Shaluf et al. [42] illustrate the problem using the case of the chemical plant in Bhopal: the company owner reported an 'incident'; the government of India qualified it as 'accident'; the affected people called it 'disaster'; and the social activists titled it as 'tragedy,' 'massacre,' and 'industrial genocide,' while Shrivastava [43] qualified the case as a 'crisis.'

'Disaster' differs from 'incident' with its broader impacts on the communities. In the EU definition, "'disaster' means any situation which has or may have a severe impact on people, the environment, or property, including cultural heritage" [44]. As the designation of 'severe' may be based on expectations, direct observations or perceptions, the European Commission provides guidelines: the national risk identification would need to consider at least all significant hazards that "would occur on average once or more every 100 years (i.e., annual probability of 1 % or more) and for which the consequences represent significant potential impacts, i.e., number of affected people greater than 50, economic and environmental costs above 100 million, and political/social impact considered significant or very serious" [45].

As a concept, 'disaster' is framing the relationships between the levels of command and management. When the capacities of the lower levels are not sufficient to cope with the consequences, the engagement of the upper levels might be necessary. The decision on the actual engagement is based on objective criteria (scope and damage), the leadership's perceptions, or the intention to secure that the disaster would not escalate into a crisis due to mass perceptions of the 'coming' threat or its consequences. Another, more conceptual explanation is provided by UNISDR: a disaster is, "a serious disruption of the functioning of a community or a society involving widespread human, material, economic or environmental losses and impacts, which exceeds the ability of the affected community or society to cope using its own resources" [27].

'Catastrophe' differs from disaster on all critical components—the scale of damage, scope of effects, level of impact, and required organisation and capabilities for preparation, response, and recovery. Compared to 'incident' and 'disaster,' it is less discussed and conceptually developed in studies and documents. However, as by presumption any catastrophe may lead to a crisis, it provides an essential guide to the CM taxonomy. From this point of view, a catastrophe's conceptual characteristics might be summarised as follows:

- "A catastrophe is defined by the magnitude of the event on an area, the capacity and ability to respond, and the time to recover" (National Homeland Security Consortium Meeting, December 2005, quoted in [46]);
- "There is a fundamental difference in the preparation, complexity, quality of effort, and scope of catastrophic disaster as opposed to a major natural disaster" [47];
- "Most or all of the community-built structure is heavily impacted" [48];
- "In a catastrophe, most if not all places of work, recreation, worship and education such as schools totally shut down and the lifeline infrastructures is so badly disrupted that there will be stoppages or extensive shortages of electricity, water, mail or phone services as well as other means of communication and transportation" [48];
- "Most, if not all, of the normal, everyday community functions are sharply and simultaneously interrupted" [25];
- "Local officials are unable to undertake their usual work role, and this often extends into the recovery period" [48];
- "... the response and recovery capabilities needed during a catastrophic event differ significantly from those required to respond to and recover from a 'normal disaster'" [49];
- "A catastrophe, however, overwhelms state and local governments and requires a federal response that anticipates needs instead of waiting for requests from below" [50];
- "Help from nearby communities cannot be provided" [48];

- "In catastrophes, compared to disasters, the mass media differ in certain important aspects. There is much more and longer coverage by national mass media. This is partly because local coverage is reduced if not totally down or out" [48];
- "In catastrophes, there is a need for a more agile, adaptable and creative emergency management" [51].

'Crisis' in the taxonomy of functions refers to communities, jurisdictions, and organisations " . . . being significantly damaged by an event or being unable to respond at maximum coherence or effectiveness when the stakes for doing so are at their highest because of a lack of resources, planning, leadership, or capacity" [52]. Farazmand [53] elaborates further:

> Crises involve events and processes that carry severe threat, uncertainty, an unknown outcome, and urgency . . . Most crises have trigger points so critical as to leave historical marks on nations, groups, and individual lives. Crises are historical points of reference, distinguishing between the past and the present . . . Crises consist of a 'short chain of events that destroy or drastically weaken' a condition of equilibrium and the effectiveness of a system or regime within a period of days, weeks, or hours rather than years . . . Surprises characterize the dynamics of crisis situations . . . Some crises are processes of events leading to a level of criticality or degree of intensity generally out of control.

Despite that reviewed sources do not connect catastrophe to crisis directly, there is an obvious link, mostly due to the anticipated almost total inability of local authorities and people to manage the situation, the interruption of most of the vital functions and services, and the psychosocial impact on the people that a totally damaged living environment has.

### 2.2.4. Management-Based Concepts: Incident, Disaster, and Crisis Management

The management-based concepts provide a framework for an effective and timely response to emergencies, including those that might be unusual or unexpected. Having the response function as a management pillar, various activities such as systematic planning, capability building, risk assessment and management, mitigation, prevention, and protection, as well as relief and recovery might be organised. As the taxonomy is to help better classify and search for tools and solutions, a relevant approach is to look at 'management' as a process that involves a number of functions such as planning, organising, staffing, capabilities development, training, directing, and consolidating communities' efforts to prepare, protect, being able to respond to and recover from the effects of natural and manmade hazards on life, property, and environment.

From the management point of view, incident, disaster, and crisis have a common feature—they all are a kind of emergencies as they contain an actual threat to the communities and their properties. In this context, 'emergency' is a "sudden, urgent, usually unexpected occurrence or event requiring immediate action" [54]. Emergency situations—and the decisions and operations to manage them—are different from the routine public protection work; nevertheless, as they are anticipated, various contingency planning, organisational, and preparation measures could be undertaken to provide relevant, effective, and timely assistance.

Presumably, incident, disaster, and crisis management should be seen as levels of an integrated approach in which every consecutive level includes and expands on the previous. Many common components might be applied across the triple scale—alert system, preparatory functions, equipment, resources, etc. However, as the size of the event matters, there is a reason for some nations and organisations to differentiate the management at '3+' levels: (1) incident management (2) disaster or emergency management; and (3) crisis management, adding when necessary specific levels, e.g., extreme events or catastrophes [55,56].

*Incident management*: the concept of incident varies depending on the way countries build the response systems. In the U.S., the National Incident Management System is " . . . for managing incidents that range from the serious but purely local to large-scale terrorist attacks or catastrophic natural

disasters," where a 'catastrophic incident' is defined as "any natural or manmade incident, including terrorism that results in extraordinary levels of mass casualties, damage, or disruption severely affecting the population, infrastructure, environment, economy, national morale, or government functions" [40]. ISO 22399 provides a more universally applicable notion: if not managed properly, the incident " … might be, or could lead to, an operational interruption, disruption, loss, emergency or crisis" [54].

*Disaster or emergency management*: presumably, the concept of disaster management should apply to the disaster-level events. Catherine Romano contends that disaster management includes "(1) preparedness planning to assess hazard vulnerability; (2) mitigation activities to reduce hazards in the structure of the facility, its equipment, its operations, and its personnel; (3) response planning to provide for key support operations, such as first aid, search and rescue, building evacuation, emergency communications, and general personnel training; and (4) recovery, in which an organization prioritizes its operations for efficient business continuation and determines how to protect and restore these components" [57].

The UN uses the concept of 'emergency management' as a universal approach, not-attached to the level of intensity of an event, " … to engage and guide the efforts of government, non-government, voluntary and private agencies in comprehensive and coordinated ways to respond to the entire spectrum of emergency needs" [27]. Salient, in this definition, is the integration of many different organisations, each with its own standard operating procedures (SOPs), towards a common goal in a complex situation.

*Crisis management*: as already stated, every crisis is an emergency, and from this point of view, the respective management concepts should not differ. The emergency planning component of management is about preparing for anticipated situations in terms of origin, localisation, and intensity. The managers are more or less aware of the possible developments, and the emergency character of a situation is determined most often by its surprise appearance or unexpectedly quick escalation.

However, the crisis situation (as well as the catastrophe) is different. A crisis starts with the unusual and unexpected. Even though entirely new types of crises are rare, a crisis may result from the surprise or uncertainty of a rapidly escalating situation. A crisis may also be born by a complex situation (simultaneous appearance of different hazards, aggravated by social reactions, etc.), significant collateral damage and secondary effects of what seems to be a 'known' emergency [52]. From this point of view, the crisis management should prepare the community and organisations for both the expected and the unusual. As Charles Hermann has underlined during the early years of the discipline, " … the term crisis has long been used to describe an event or problem that "(1) threatens high-priority values of the organization, (2) presents a restricted amount of time in which a response can be made, and (3) is unexpected or unanticipated by the organization" ([58], as quoted in [52]).

Building crisis management capacity, organisation, skills, resources, and culture is about preparing to cope with extraordinary and unexpected situations to save lives and property. From this perspective, the taxonomy reflects three notions of 'crisis management.

*'Narrow' crisis management*: in a narrow context, the crisis is seen through the prism of decision-maker(s) as " … a situation that threatens high priority goals of the decision-making unit, restricts the amount of time available for response before the decision is transformed and surprises the members of the decision-making unit by its occurrence" [59]. The 'narrow' crisis management is a response rather than planning, organising, mitigating, preparing, etc. The response is provided in a situation, which is unexpected or planning has not been made due to a variety of reasons, or the crisis management organisation has been made dysfunctional by the event. As the crisis profoundly affects the way of life and functioning of the community, the 'narrow' crisis management might also be seen as change management [52].

*Comprehensive crisis management*: to meet the specific requirements to the CM taxonomy, the 'traditional' view of emergency management 'phases,' introduced by the National Governors' Association' guide in 1979 [60], are expanded to integrate hazard awareness and community vulnerabilities into a better understanding of complex contemporary risks, improvements of

the governance frameworks and mechanisms coverage on the entire spectrum between hazards identification, mitigation, preparation, protection, response, and recovery, rehabilitation and reconstruction till building reliable and sustainable community resilience. The management of every functional area is based on the Henry Fayol's classical formula "to manage is to forecast and to plan, to organise, to command, to coordinate and to control," expanded with: building common goals across the various crisis management stakeholders, motivating the people for volunteerism and mutual aid, coordinating across a large number of very different organisations engaged in crisis management, sense-making, building meaning through comprehensive communications, etc. Briefly, comprehensive management is about all hazards, all public security stakeholders, all systems, all resources, and all actors across all jurisdictions.

*Crisis management function*: the crisis management function aims at achieving effects, e.g., coordination, a direction of effort, shared awareness, etc., in a crisis management system of systems. The 'function' focuses on what is to be achieved, not how or by whom. Several systems, tools, building blocks, etc., may individually or in concert deliver a given function and conversely, they may support several different functions [61].

As the taxonomy encompasses the central, regional, and local levels, 'emergency management' is applied to situations in which the affected communities have the capacity to cope. 'Disaster management' is about situations in which the affected communities might need and require support from upper-level jurisdictions. The comprehensive and narrow approaches are used to combine the preparation (mitigation, capability development, and strategic adaptiveness) with the response (response, relief, and recovery) aspects of the crisis management to be able to cope with both expected emergencies and surprising developments in terms of size, scope, destructiveness, etc. As some authors have summarised, crisis management may either build on emergency management plans and protocols with auxiliary resources, or it may augment the more rigid, but efficient, decision-making and problem-solving protocols with selectively used practices that help responders adapt to specific disruptions to the emergency response infrastructure [62,63].

*2.3. Conceptual Model*

The taxonomy model is a systematic conceptual representation of a crisis management system existence [11]. Accordingly, the main functional areas, functions, and tasks performed by different authorities, organisations, and individuals within the comprehensive crisis management framework are established on the principles of the system theory in their hierarchical relationships. The model is built with a twofold objective: (1) to reveal the comprehensive approach to the modern crisis management (in the scope defined above); and (2) to meet the needs for the proper, easy-to-use, and sustainable classification both of practitioners and solution providers. From this point of view, the model serves the building of the POS platform and provides opportunities for these solutions to be found, tested, evaluated, used, and enriched by national and European crisis managers, planners, and researchers.

The CM taxonomy is a multi-dimensional hierarchical classification of crisis management functions within the civil protection mission and in the internal security ('homeland' in the U.S. parlance) context. This framework derives from both the historical evolution of the civil protection from the traditional 'civil defence' [64,65] and the current European Union understanding of civil protection as co-ordinated, effective, and efficient response to natural hazards and manmade threats. For example, the objective set by the European Union Internal Security Strategy to increase Europe's resilience to crises and disasters [66], which is reflected in the relevant policy documents and legal acts of member states, determines the particularly broad scope of the CM taxonomy.

2.3.1. Basic Assumptions

According to the European Parliament and the Council,

In view of the significant increase in the numbers and severity of natural and manmade disasters in recent years and in a situation where future disasters will be more extreme and

more complex with far-reaching and longer-term consequences as a result, in particular, of climate change and the potential interaction between several natural and technological hazards, an integrated approach to disaster management is increasingly important. [44]

Characteristics such as the complexity, cross-sectoral impact of hazards, combination of hazards and malicious acts, and highly dynamic escalation call for enhancing crisis management practice in terms of effectiveness, efficiency, and strategic adaptiveness. Responding to these realities and perspectives, the CM taxonomy is built on the following assumptions:

- The scope and impact of natural and manmade hazards and threats to European communities evolve and challenge the European collective crisis management mechanism and national capabilities;
- The importance of the civil protection function as a component of the European and national internal security is growing;
- There is a growing need, as well as willingness for joint operations for crisis management and disaster resilience [67];
- Crisis management at the European and national level needs better evidence-based investment decision making for building a well balanced comprehensive Portfolio of Solutions in terms of tools, operational concepts, and approaches.

### 2.3.2. Approach

To cope with the challenge of complexity and comprehensiveness, the CM taxonomy goes beyond the popular 'phased' model of crisis or emergency management that includes mitigation, protection, response, and recovery [68]. This is done to avoid the impression that the phases are somehow independent and discrete, that the end of the previous means the begining of the next, etc., and that the taxonomy of every phase is a facet of the overall taxonomy. The review of various research works underlines that the so-called 'crisis management phases' should not be seen as linear. According to Boin et al., "Linear thinking ('big events must have big causes') has given way to a more subtle perspective that emphasizes the unintended consequences of increased complexity. Crises, then, are the result of multiple causes, which interact over time to produce a threat with devastating potential" [69]. An early eminent report emphasises the cyclical relationships among these four phases of disaster activities, illustrating how the recovery efforts ("like using loans to relocate residents out of floodplain") may have a mitigation effect against future disasters [68]. David Neal's comprehensive study proposes, among others, approaches to re-examine the disaster (management) phases with the understanding that they are mutually inclusive, multi-dimensional, and should be seen not only through the organisational prism of the crisis managers, but also from the perspectives of responders and the affected people [70].

Reflecting such methodological considerations, this conceptual model encompasses through taxonomy:

- Citizens, private subjects, and public authorities;
- Strategy, policy, and operations;
- Missions, functions, and tasks;
- Organisations, processes, and activities;
- Human as well material resources, and real estate;
- Command, control, coordination, management, etc.

The taxonomy provides opportunities to see how different crisis management entities (leaders, organisations, people, resources, information, etc.) are involved in policy and efforts to reduce the risks of hazards, to build effective and sustainable capabilities, to strengthen the strategic adaptiveness of the system, to provide day-to-day protection, to be able to respond when the crisis goes beyond incident and even disaster and to provide relief and recovery, making the life of the affected people even safer and better than before. Potentially, it covers a large timeframe in which societies are expected

to pass through different stages of maturity and ability to cope with natural and manmade hazards, building the quality and culture of resilience.

In terms of usage, the taxonomy meets the requirements of both crisis management practitioners and solutions providers. For that purpose, the classification is designed following the top-down approach from crisis management decision-makers towards the organisations, responders, and citizens. On the other hand, solution providers may see the crisis management functions decomposed to tasks, operations, and activities and to allocate envisioned and developed products to one or more of them, where the solution fits best.

### 2.3.3. Context

The conceptual model of the CM taxonomy reflects the socio-political contexts within which the crisis management takes place. This is the *internal security* in the European Union, that sets up the crisis management of hazards, or the *homeland security* in the US. Putting aside other conceptual issues, the taxonomy of crisis management functions reflects the overall internal security taxonomy. The aim is to avoid, broadly said, an interoperability gap that might arise in critical situations in which different leaders, organisations, and people act with a different understanding of the facts, signals, and commands, and follow different procedures and communication languages. From this point of view, the conceptual model is context aware.

### 2.3.4. Multi-Dimensionality

The taxonomy model's multi-dimensionality reflects the applied comprehensive understanding of crisis management, its command and management architecture and the engagement of people, societal, and private actors, and the set of standardised processes and procedures for setting up policy and strategy, planning and organising, capacity building and exploiting.

'Function' is the core of this taxonomy model. Taken at the upper level of understanding, a 'function' is an established-by-design characteristic or ability to realise processes of certain sorts for gaining particular outcomes or meeting concrete objectives [71]. It is "the action for which a person or thing is specially designed, fitted, used or intended to accomplish or execute" [72]. From the public governance point of view, the crisis management 'function' is a high-level purpose, responsibility, task, activity, and transaction assigned to a particular authority, organisation, as well as to the citizens, by legislation, mandate or policy [73].

Functions are related to purpose (e.g., 'protection') and processes (e.g., 'planning,' 'organising,' 'training,' etc.), explaining what and why should be done and in which way. The approach follows one of the well established models with five logical steps: (1) signal detection, (2) preparation/ prevention, (3) containment–damage limitation, (4) recovery, and (5) learning [74].

Applying such logic, the mission (crisis management) is realised through three functional areas: 'preparatory,' 'operational,' and 'common.' Such an approach has been used in the project ACRIMAS to define the capability requirements for managing the aftermath of crises and identify clusters of managers' practical needs [75]. The ResiStand Consortium has also used the model for an EU-sponsored study on increasing disaster resilience by the standardisation of technologies and services [67]. *Preparatory* are those functions aiming to reduce the loss of life and property by reducing the potential impact of natural and manmade hazards, to shape protection, response, and recovery capabilities, and to build strategic (long-term) crisis management adaptiveness. *Operational* are the functions aimed at the day-to-day protection of people and infrastructure, relevant response to crises, relief, and recovery from the consequences of a crisis. The *common* functions provide critical support for decision making and operations throughout the crisis management spectrum. Figure 2 illustrates this categorisation of crisis management functions.

Furthermore, the taxonomy model's 'key functions' are established on Henri Fayol's core management elements of planning, organising, command, coordination, and control [76], expanded and adapted to the preparatory, operational, and common functional areas specifics. To make the

taxonomy easier to use by crisis management practitioners, some of the key functions are 'horizontally' expanded, e.g., in the 'capability development' functional area, 'organise for crisis management' is expanded with 'establish CM doctrine and train organisations and people'; in the 'response' functional area, the implementation function is explained by 'respond to the hazard,' 'limit the impact of the crisis,' and 'support affected people'.

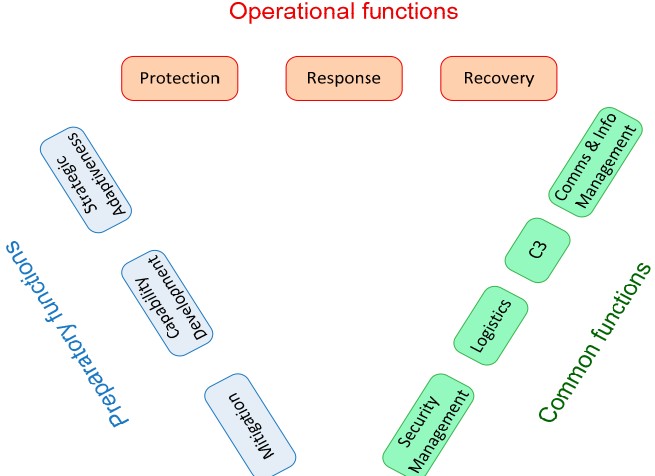

**Figure 2.** Functional areas of the crisis management taxonomy.

The third and fourth taxonomy levels (sub-functions and tasks) are defined through decomposition of functions in accordance with a modified and adapted 'OODA loop'—observe, orient, decide, act—proposed by John Boyd [77,78]. The model is selected as proven in practice for dealing with the volatility, uncertainty, complexity, and ambiguity (VUCA) of crises (Figure 3). The four elements of the 'loop' have different interpretations throughout the preparatory, operational, and common functional areas. They reflect the reality that, during the preparatory period, the loop is 'slow' and to some extent, the management gradually transits into a long-term policy (e.g., for strengthening resilience), while during the 'response,' the 'OODA loop' is expected to be more rapid and leadership driven.

The complete set of crisis management functions 'connects' hazards and emergencies with communities' security. The way we think about hazards/emergencies, on one side, and the exposed communities on the other, determines where the search for solutions should be directed. To cope with this challenge, the taxonomy builds on nine complex elements that determine the scope and quality of the crisis management system, as illustrated in Figure 4: legal norms and procedures; policy and strategy; authorities, organisations, and citizens; information, awareness, and knowledge; plans, programmes, and resources; doctrine, training, and skills; equipment, infrastructure, and networks; guidelines, commands, and management; operations, services, and measures. The taxonomy facilitates further harmonisation across these elements to make the crisis management system synchronised and inclusive.

The model's physical application dimensions include four principal levels of decision making, organising, planning, training, and acting that need to be defined for each particular country-specific case:

- National (also central, federal or state) level;
- Regional (also a province, governorate or wider specific geographical area);
- Local (also community or an administrative entity at the level below province/governorate);
- Cross-border (also European or international).

In addition, the crisis might be localised or cover a wide area. The localised crisis has a clearly identifiable scene. A wide-area crisis can be generated by connected acts at multiple sites in a wide

area, as in the September 11, 2001 terrorist attacks on the US, or when wide areas are affected to some degree, e.g., by widespread flooding, a pandemic, sustained power outages or severe weather [79].

Decision making, command, coordination, and supporting functions such as communications, information management, logistics, and the provision of security are common for all preparatory and response functions. They are not 'less important' or of a secondary priority, and form the backbone of the crisis management continuum. Figure 5 illustrates the multi-dimensionality of the taxonomy model.

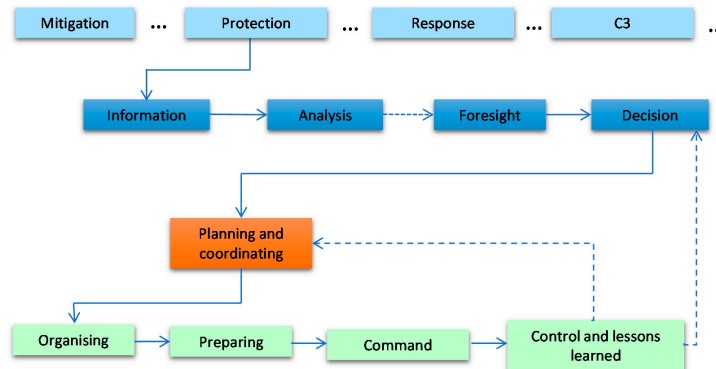

**Figure 3.** Illustration of the decomposition of the crisis management functions.

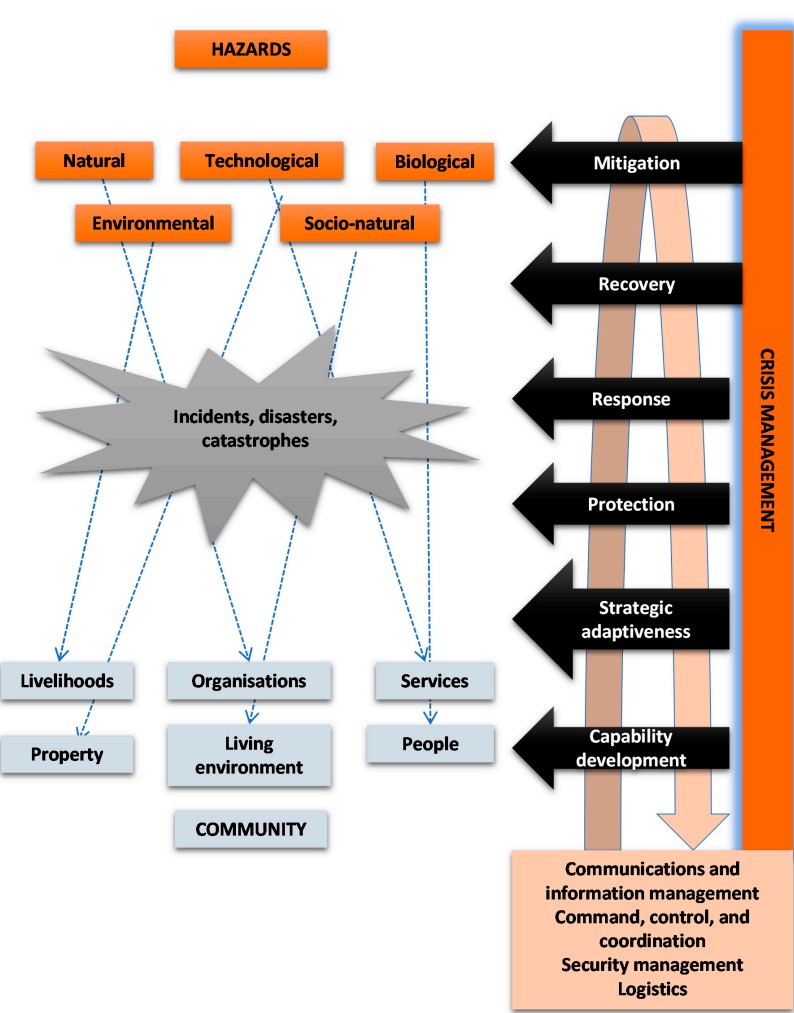

**Figure 4.** Key elements of the crisis management system in hazard-community context.

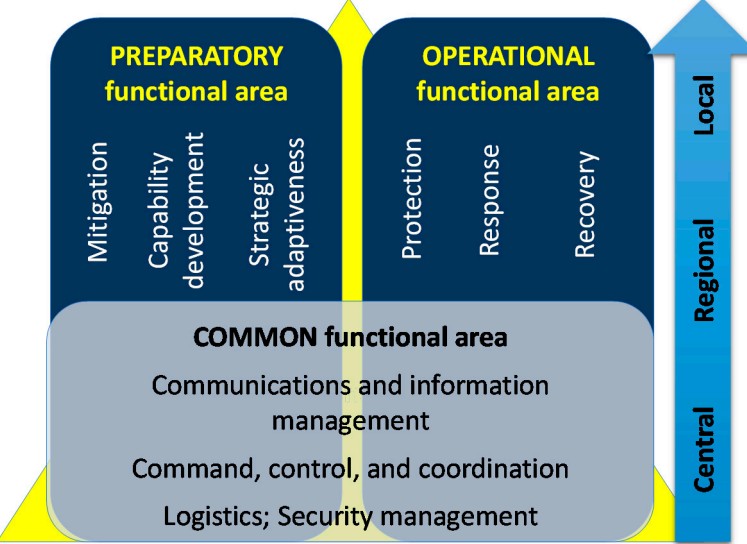

**Figure 5.** Illustration of the taxonomy dimensions.

### 2.3.5. Language

In terms of the language, the CM taxonomy is developed using the professional language of crisis management practitioners. Words (e.g., responder, volunteer) and standardised phrases (e.g., common operational picture) are taken from normative documents, guides, glossaries of terms, plans, etc. Thus, the model is made easy to use for the practitioners, while the solution providers are expected to invest the necessary effort to be able to understand and use this language.

### 2.3.6. Commonality

Not only the language but also the selected structuring of the functional areas of the CM taxonomy corresponds to widely used classification schemes. Figure 6 presents the correspondence with the UN Sendai Framework for Action [1], the classification used by the UN Office for the Coordination of Humanitarian Affairs, or OCHA [80], the EU Civil Protection mechanism with the Emergency Response Coordination Centre (ERCC) [44], the Universal Task List (UTL), used by the U.S. Department of Homeland Security [81], the organisation of critical infrastructure protection in EU Directive 2008/114 on critical infrastructure protection (CIP) [82], the framework for providing the cybersecurity of critical infrastructures, developed by the U.S. National Institute of Standards and Technology (NIST) [83], and two relevant research projects—ResiStand [84] and S-HELP [85]. As illustrated in the last column in Figure 6, two of these classification schemes—of UN OCHA and the UTL of the Department of Homeland Security—cover issues beyond the scope of the CM taxonomy. Most importantly, there is either direct correspondence between the first-level categories or easily established relations with and between lower-level taxonomy fields.

| | Preparatory functions | | | Operational functions | | | Common functions | | | | |
|---|---|---|---|---|---|---|---|---|---|---|---|
| CM Taxonomy | Mitigation | Capability Development | Strategic Adaptiveness | Protection | Response | Recovery | Crisis C&I Management | C3 | Logistics | Security Management | |
| UN/Sendai[1] | Mitigation (understand & reduce risk) | Preparedness | Knowledge; cooperation; Innovation & technology; resilience | Prevention | Response | Recovery and Rehabilitation | | | | | |
| UN/OCHA[80] | Preparedness & risk management | | Community engagement | Protection | Response | | Information Management | Coordination | Logistics | | Humanitarian Development Nexus, Financing |
| EU 1313[44] | Prevention | Preparedness Readiness | | | Response | Relief | | ERCC | Logistical Support | | |
| DHS UTL[81] | Protect | Preparedness Supporting Tehology | | Protect | Respond | Recover | Comms & Info Management | | Resource Management | Prevent (access control) | Prevent (Counter-Terrorism; Intel) |
| CIP EU 114[82] | Risk analysis Measures | Measures | | Protection | Graduated response | | Communications | | | Security of info systems; access control | |
| NIST Cyber[83] | Identify | | | Protect Detect | Respond | Recover | | | | | |
| ResiStand[84] | Preparedness / Risk assessment Exposure reduction | Capacity Development | Trend analysis Monitoring & Review | Monitoring and Detection | Response | Recovery | Sit Assessment Information Management (in Response) | C3 (in Response) | Logistics (in Response) | Security/ Law Enforcement (in Response) | |
| S-HELP[85] | Mitigation | Preparedness | | | Response | Recovery | | | | | |

**Figure 6.** Comparison of the taxonomy's functional areas with other classification schemes.

### 2.3.7. Adaptability

The functional character of the model provides good opportunities to further the taxonomy's precision, to develop and adapt it with the accumulation of knowledge and changing circumstances. Any consequently developed solution or tool could be applied to an existing or newly established function, sub-function, task, and activity. That would not change the taxonomy construct; rather, it will improve its consistency and usefulness.

## 3. Taxonomy of Crisis Management Functions

The CM taxonomy is developed in three broad functional areas—preparatory, operational, and common (see Figure 2). Every functional area includes clusters of key functions, sub-functions, and tasks. The preparatory functions include mitigation, capability development, and strategic adaptation, with 'community resilience' being part of the latter. The operational functions include protection (the day-to-day civil protection operations and measures), response, and recovery, which is sub-divided into immediate relief and long-term recovery and reconstruction. The Common functions are those that support most of the other functions providing crisis communications and information management (CCIM), command, control, and coordination (C3), logistics, and security management (for the complete taxonomy refer to the Supplementary Material to this article).

This section explains the rationale behind the choices in the design of the taxonomy and provides an illustrative visual representation of one of the functional areas. The supplement provides a detailed description of the functions, sub-functions, and tasks included in the taxonomy.

### 3.1. Preparatory Functional Areas

The preparatory functional areas reflect the guidelines formulated by the Sendai Framework of Disaster Risk Reduction 2015–2030, outlining four priorities for action:

- Understanding disaster risk (in all its dimensions);
- Strengthening disaster risk governance to manage disaster risk;
- Investing in disaster risk reduction for resilience;
- Enhancing disaster preparedness for effective response and to "Build Back Better" in recovery, rehabilitation, and reconstruction [1].

The functions defined here as *preparatory* also reflect the EU Parliament and the Council Decision 1313 for establishing the Union Civil Protection Mechanism. The Mechanism is established as a " . . . general policy framework for Union actions on disaster risk prevention, aimed at achieving a higher level of protection and resilience against disasters by preventing or reducing their effects and by fostering a culture of prevention, including due consideration of the likely impacts of climate" [44]. The quoted EU document focuses primarily on response capabilities and readiness for action, defining 'preparedness' as:

> a state of readiness and capability of human and material means, structures, communities and organisations enabling them to ensure an effective rapid response to a disaster, obtained as a result of action taken in advance. [44]

ISO 22300 reflects a similar position, defining 'incident preparedness' as "activities taken in order to prepare incident response" [26]. UNISDR's understanding is more comprehensive also for the notion of 'readiness' [27]. Blanchard has identified 34 definitions of 'preparedness' by overviewing just U.S. official documents and supporting sources [46].

'Readiness' to act is the essence of 'preparation,' but it is only one of the comprehensive crisis management aspects. The CM taxonomy elaborates on the approach of the International Federation of Red Cross and Red Crescent Societies that was found to be closer to the comprehensive nature of crisis management. The preparatory functional areas are defined with the broader aim to mitigate the sources of threats; to reduce exposure and vulnerabilities; to predict and where possible to prevent disasters and crises; to provide response capabilities that are effective and maintained at the relevant level of readiness, capabilities to cope with the immediate disaster consequences and to provide recovery and reconstruction; and to contribute to building comprehensive and sustainable resilience capacity throughout the society, communities, and businesses.

The functional areas are generally organised following the ISO 22301 "'Plan–Do–Check–Act' (PDCA) cycle to planning, establishing, implementing, operating, monitoring, reviewing, exercising, maintaining and continually improving the effectiveness of an organization's BCMS" (Business Continuity Management System) [86].

The first two functional areas—mitigation and capability development—cover the functions aimed at understanding risks, reducing exposure and vulnerabilities, and developing crisis management policies and a broad set of capabilities. The third one—strategic adaptiveness—reflects on concepts of agility, anticipation (foresight), and adaptiveness of the crisis management arrangements to significant changes in the environment, e.g., in hazards/threats, technologies, societal expectations, and building and measuring community resilience.

### 3.1.1. Mitigation

Mitigation is the functional area on which the concepts of 'hazards' and 'vulnerability' meet and, if not managed properly, may trigger a 'crisis.' From this point of view, the U.S. Federal Emergency Management Agency (FEMA) defines mitigation as "the effort to reduce loss of life and property by lessening the impact of disasters" [87]. ISO 22320 describes mitigation as the "measures taken to prevent, limit and reduce impact of the negative consequences of incidents, emergencies and disasters" [88]. This is achievable through conducting hazard-specific identification, tracking, and assessment, as well as mapping and the permanent monitoring of their conditions and possible escalation. Consecutively, a vulnerability assessment methodology is applied to estimate the possible impact of concrete hazards on concrete communities, peoples' lifelines, and critical infrastructure.

The assessment stage culminates in risk evaluation. A detailed risk assessment is based on the impact's intensity and the likelihood of occurrence of different hazards. Risk evaluation (following ISO 31000) "involves comparing the level of risk found during the analysis process with risk criteria established when the context was considered" and supports making decisions that "take account of

the wider context of the risk and include consideration of the tolerance of the risks borne by parties other than the organization that benefits from the risk" [89].

It is important to note that the mitigation functional area is established with the understanding that the complete prevention of losses is ultimately unattainable, and a mitigation effect on the plausible hazards could be achieved only in a long-term perspective. The taxonomy directs the search of solutions towards the concrete vulnerable assets threatened by concrete hazards, and this is used to identify tasks, actions, and other measures to reduce damages [36].

On that basis, crisis management organisations set long-term goals and a strategy, develop objectives and programmes, allocate resources and initiate organisational and legal measures, and take action at the national, regional, and community levels. The overall goal is to mitigate risks from hazards in a manner relevant to potentially affected people, properties, local infrastructure, critical infrastructures, and critical state functions. Concretely, ISO 22301 recommends " . . . measures that: reduce the likelihood of disruption; shorten the period of disruption; and limit the impact of disruption on the organization's key products and services" [86]. UNISDR recommends that "mitigation measures encompass engineering techniques and hazard-resistant construction as well as improved environmental policies and public awareness" [27]. Particular focus is placed on shared information, education, and the training of all potentially affected communities and organisations [67,89].

The proposed cluster of functions is expected to produce a mitigation effect on communities and the state in a long-term perspective. FEMA has emphasised mitigation as the most effective and cost-efficient strategy for dealing with hazards. Its implementation requires a comprehensive and sustainable strategy-based approach, including monitoring, evaluation, and measuring for keeping the strategy relevant. In some aspects, the mitigation action taken in advance might contribute to the prevention of and thus to completely avoid potential adverse impacts [27].

Implementing the mitigation functions also provides information for comprehensive capability development and planning the strategic adaptiveness of the crisis management system. Hazard mitigation is an element of building resilience throughout communities, nationally, and regionally.

### 3.1.2. Capability Development

Capability development functions address the overall crisis management system and the potentially affected population. Every organisation and everyone who might be a responder of a certain kind or a victim in a crisis need to take measures for better organisation, information, preparation, and action. People, equipment, and resources are integrated into an effective mechanism through crisis management doctrine, organisation, planning, training, command, control, and coordination. As UNISDR explains, "Preparedness action is carried out within the context of disaster risk management and aims to build the capacities needed to efficiently manage all types of emergencies and achieve orderly transitions from response through to sustained recovery" [27].

Capability planning is a method created to be relevant for domains with a high level of volatility, uncertainty, complexity, and ambiguity (VUCA), such as crisis management. Tagarev defines the term 'capability' as " . . . the capacity, provided by a set of resources and abilities, to achieve a measurable result in performing a task under specified conditions and to specific performance standards" [90]. Paul K. Davis, a renowned author on the issue, argues that the method is about "planning, under uncertainty, to provide capabilities suitable for a wide range of modern-day challenges and circumstances while working within an economic framework that necessitates choice" [91]. ISO 22315 introduces the term as an element of the disaster management preparatory function: "Knowledge and capacities developed to effectively anticipate, respond to, and recover from the impact of likely imminent or current hazard events or conditions" [92].

To establish a proper ground for the application of capability planning, the taxonomy elaborates the functions–sub-functions–tasks relationships to involve a functional analysis of operational requirements. In this context, the major building blocks are established using proven practices in the field of defence.

First, high-level guidance is provided through the capability development policy framework. It reflects the comprehensive character of the goal to build capabilities to act effectively in planned and unexpected crises using different organisations; national, regional, and local authorities; public and private entities; communities and individuals; professional responders and volunteers; as well as a variety of resources. Second, the taxonomy envisions the development of a crisis management doctrine as an operational platform on which different agencies, responders, and affected people will act and look for assistance. Third, the taxonomy applies the system of systems approach to group different functions as sources of capability clusters (or capability partitions). The systems encompass the core imperatives of crisis management, such as doctrine, organisation, human resources, equipment in operational systems, infrastructure, education and training, learning lessons, etc. The capability method recognises the interdependence between these imperatives. Fourth, it uses scenarios to identify the most effective combination of capabilities and, respectively, the optimum development investments across different agencies and authorities [93].

Capabilities are of high value only if a high level of functional and technical interoperability is built in in the process of design and acquisition. Interoperability is seen as a critical source of gaps throughout the crisis management system.

Capability development is a recurring function. A set of indicators are established for the identification, analysis, and evaluation of the key preparedness features as comprehensive awareness, systems and organisations capacities, readiness for response and relief operations, etc. [27]. EU projects, such as DRIVER+, provide for trialing capabilities, sharing information on performance in various circumstances, exploring organisational and technological issues, and thus facilitating innovation. Capability development for crisis management supposes systematic support by formal institutional, research, technological, legal, and budgetary capacities [27].

### 3.1.3. Strategic Adaptiveness

The concept of 'strategic adaptiveness' in the hazard-oriented crisis management domain reflects the shared understanding between scientists and practitioners about the limits of linear, deterministic approaches and predictive models in the field of civil protection [94,95]. Responding to similar observations, Carl Walters appeals that "Instead of seeking precise predictions of future conditions, adaptive management recognizes the uncertainties associated with forecasting future outcomes, and calls for consideration of a range of possible future outcomes" [96]. Hazards, both natural and manmade, that drive the crisis management function, are complex and dynamic, and change and surprise are their essential features [97]. Civil protection itself is an adaptive system and a function. Its performance might be improved by continuously adjusting organisations, legal frameworks, resources, systems, doctrine, and operations as a reflection to the environmental monitoring, analysis, experimentations, lessons learned, foresight, etc., performed across the civil protection domain.

The functional area 'Strategic adaptiveness' is included in the taxonomy with the aim to prompt the civil protection decision makers, stakeholders, managers, and people " . . . to recognize the limits of knowledge and the need to act on imperfect information" [98]. Adaptiveness is achieved by performing a cluster of management functions reflecting the evolutionary character of three key components: the ecosystem, the societal system, and the technology systems used to help communities to cope with negative natural and manmade situations. Changes are identified through monitoring, analysis, and foresight, and findings are turned into decisions to invest in a capacity to adapt in a timely manner. Adaptation may be 'reactive' or 'active,' in the latter case, relying on 'alternative futures' scenarios [99–101] and incorporating developments in advance of anticipated changes. For example:

- Strategic adaptiveness to the ecosystem evolution may define three basic tasks: adaptation to gradual changes that may lead to the escalation of known hazards, the adaptation of measures to reduce the risk of hazards that may rise to extreme levels, and adaptation to the geospatial change of hazards towards regions, which previously have not been threatened [102];

- Strategic adaptation to changes in the societal system is much broader and entails complex structural domains as demographics, psychosocial developments, urbanisation, volunteerism, governance, and many others;
- The strategic adaptation tasks may include the reorganisation of the crisis management system towards more decentralisation or centralisation; the expansion of volunteers and public–private formats or strengthening the professional corps of responders; building an extended capacity to provide mental health and psychosocial support; the introduction of different methods of emergency sheltering; etc. Strategic adaptation to technology evolution used in crisis management may include the use of off-the-shelf assets, the adaptation of commercial or military assets to the responders' special needs, or the design of advanced software tools and hardware.

In the face of uncertainty, successful are the organisations with a shared purpose, allowing the free flow of information and knowledge, able to integrate and mobilise resources, centred on knowledge, maintaining diverse and evolving competencies, facilitating networking and cooperation. Such organisations are able to shorten the learning cycle drastically, adapt, self-organise, make decisions, and take actions [103]. Having such qualities is essential for each individual organisation, as well—and even more importantly—for the network of public, private, and societal organisations contributing to crisis management.

The degree to which the strategic adaptiveness function positively affects the decision-making and implementation process across all the preparatory, operational, and common functions determines its significance in practice.

Between all the discussed functions, the strategic adaptiveness seems to be the most prone to science, advanced developments, and strategic thinking. The taxonomy frames some of the most important tasks that can facilitate the development of innovative tools and solutions in terms of scientific and learning exercises (e.g., the development of alternative hazards and crisis operations models, scenarios, simulations, computer-assisted role games, etc.), advanced information management solutions especially for real-time assessment and forecasts, command control, and coordination models in a catastrophic situation; alternative organisational arrangements and processes; maintaining a stock of specific assets, e.g., vaccines, that seem relevant for unlikely scenarios (e.g., [104]), etc.

In a broader and longer perspective, the strategic adaptive management should help to build a resilience capacity across the natural and man-made ecosystems, communities, essential services, organisations, and nations. Introducing the Union Civil Protection Mechanism, the European Commission declared:

> The Union Mechanism should include a general policy framework for Union actions on disaster risk prevention, aimed at achieving a higher level of protection and resilience against disasters by preventing or reducing their effects and by fostering a culture of prevention, including due consideration of the likely impacts of climate change and the need for appropriate adaptation action. [44]

Resilience is included in the strategic adaptiveness functional area, with the understanding that it contributes to the capacity of the community to self-organise and successfully cope with the shocking events caused by natural and man-made hazards [105]. In the crisis management context, a critical resilience capacity is the ability at all levels of command and management to maintain awareness in a highly complex, dynamic, and harmful situation, to take relevant and timely decisions and to communicate them, to manage their implementation and conduct operations, and to quickly re-establish the broken vital functions for communities and nations.

As an illustration of all ten functional areas of the taxonomy, Figure 7 presents the preparatory functional area "strategic adaptiveness." The supplement to this article provides descriptions of each function, sub-function, and task.

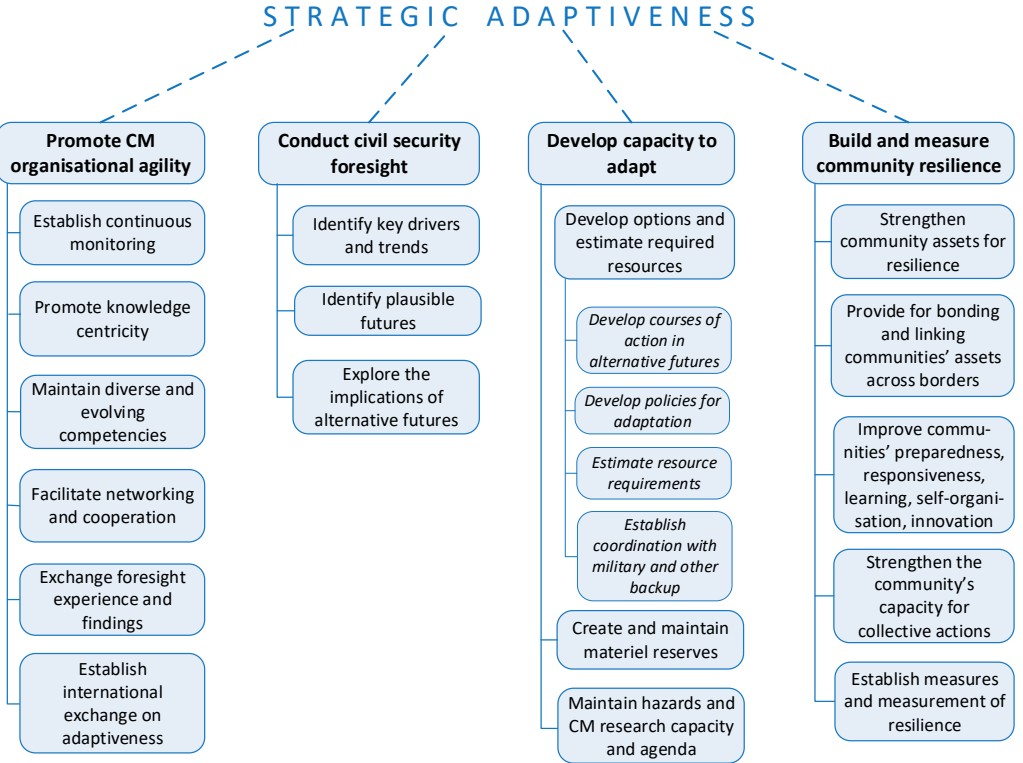

**Figure 7.** Structure of functional area "strategic ddaptiveness".

### 3.2. Operational Functional Areas

The operational functions are organised in three areas: protection, response, and recovery. The purpose of the operational functional areas is to establish an organisation and procedures for regular protection and incident management, response to major emergencies and crises that disrupt the normal community, business, and national functioning and to provide immediate relief, the complex recovery of affected people, assets, and environment, and the continuity of operations.

Making decisions and conducting operations are the essence of crisis management. They are time-sensitive, with little leeway for delay or ad hoc improvisations, might be specific or comprehensive, performed while the hazard continues, applied on different terrains, time of the day, weather, etc. Nevertheless, according the World Health Organization, "80 % of what we do in emergencies is generic, 15 % is hazard-specific, and 5 % is unique to the event" [106]. The taxonomy is designed to help allocate solutions and tools in a manner that will diminish the uniqueness of the crisis management operations by strengthening their generic character and keeping at the same time their hazard-specific effectiveness.

The EU places the emphasis of the civil protection operations on " … primarily people, but also the environment and property, including cultural heritage, against all kinds of natural and man-made disasters, including environmental disasters, marine pollution, and acute health emergencies, occurring inside or outside the Union" [44].

The classification of operations and other measures, undertaken during protection, response, and recovery, is not deterministic—protection operations may quickly turn into a response in the case of the surprising escalation of an incident; relief and recovery operations may start while the response is still undergoing; protection, response, and early recovery (relief) operations may be conducted simultaneously; etc. According to the DHS, "these overlapping areas are identified through comprehensive planning with the whole community to ensure that they are properly addressed during the response to an incident. Ensuring that operational plans properly account for the integration of mission areas is essential" [107].

### 3.2.1. Protection

The protection functional area encompasses the functions, tasks, and activities performed on a day-to-day basis to provide comprehensive civil protection against various hazards and threats that do not reach the level of 'crisis.' This is a routine function and is performed mostly by professional response organisations. Its key features are twofold. On the one hand, the protection functional area overlaps with capability development and mitigation. That means, during the process of building capabilities and reducing hazards, threats and vulnerabilities, operations to protect people, local and critical infrastructure and assets, social and industrial supply chains, etc. are performed at a level below crisis management. Protection functions are especially important if mitigation is problematic or poorly functioning [108].

The protection function is well defined by the U.S. Department of Homeland Security's National Protection Framework:

> The National Protection Framework focuses on Protection core capabilities that are applicable during both steady-state conditions and the escalated decision making and enhanced Protection operations before or during an incident and in response to elevated threat. Steady-state conditions call for routine, normal, day-to-day operations. Enhanced conditions call for augmented operations that take place during temporary periods of elevated threat, heightened alert, or during periods of incident response in support of planned special events in which additional or enhanced protection activities are needed. [107]

On the other hand, the protection function is bonded with the response and recovery operations performed in reactions to incidents and those disasters that do not escalate into a wider socio-security crisis. For solution providers, an important consideration is that at each moment, an incident or disaster may escalate into a crisis, and respectively, the response and recovery functions would be engaged.

Three of the functions in the functional area 'Protection' are elaborated with a respective focus on public protection (including health, buildings and community infrastructure safety, and mass events security), critical infrastructure protection, and critical information infrastructure protection.

### 3.2.2. Response

In the CM taxonomy, the response functional area derives from three crisis management considerations: the consequences from the hazard for the society are or could be significant; the threat is imminent and could escalate quickly; the local authorities do not have sufficient capacity to cope with the threat. This approach combines disaster, emergency, and crisis categories in one highly intensive but relatively short period of crisis management in which immediate measures and operations are undertaken "to save lives and to limit adverse effects" [67] (p. 14). As Christensen et al. accentuate, "it is important to contain a crisis or disaster in order to minimize damage and prevent essential systems from collapsing. This may be dependent on prevention and preparation" [108].

Operations are the essence of the response function. They are defined with two main goals: to limit the scope of the damage and to support the affected people. The taxonomy elaborates operational tasks across the full cycle of orientation, decision making, mobilisation of responders and resources, command of operations, and preparation for immediate relief and comprehensive recovery. It further includes two specific elements unique for the response functional area: introducing a special legal regime over a concrete affected area, or even for a whole state, for a limited period; law enforcement measures to isolate and protect the affected area and population from eventual marauding and other criminal violations.

The taxonomy pays special attention to the command and management decision making, considering the expected shortage of information, highly dynamic and hard-to-predict developments, complexity and various psychosocial impacts on the population, and the stress on the decision-makers. The tasks are defined to guarantee that alerting and special measures are taken adequately, operations are conducted to help the people first, and resources are delivered promptly. Crisis leadership during

response is framed by five critical tasks: orientation, decision making, public communication, a quick exit from the crisis towards relief and recovery, and learning from the experience [69].

The taxonomy includes here the tasks related to the provision of international support, e.g., notification for disasters within the European Union, requesting support through the Emergency Response Coordination Centre and the Union Civil Protection Mechanism, and directing assistance interventions [44].

### 3.2.3. Recovery

The recovery functional area includes the functions, sub-functions, and tasks towards returning, as soon as possible, the livelihoods and living conditions of the affected people and the functioning of organisations, businesses, infrastructure, and assets to normality [26,27,67].

The taxonomy reflects this objective from two perspectives—immediate relief in terms of vital assistance to the most affected people and long-term recovery not only to re-establish the pre-crisis health, livelihoods, and material conditions but to improve them and develop further critical services and assets. As the U.S. DHS recommends, "Coordination with the pre- and post-disaster recovery plans will ensure a resilient recovery process that takes protection into account. Protection and Mitigation focus on a sustainable economy and community resilience and not just the swift restoration of infrastructure, buildings, and services" [107] (p. 30). The long-term approach to recovery, along with enhancing the strategic adaptiveness, contributes to enhancing community, economy, and national resilience.

Adapted from FEMA's 'capstone doctrine,' the functions and tasks are defined to provide assistance to the affected people, to restore critical public services, to help make the damaged infrastructure operational, to restore the economy, and to mitigate the adverse effects on the environment [109].

### 3.3. Common Functional Areas

Four of the functional areas are defined as 'common,' i.e., they include functions and tasks that are performed to a different degree across the crisis management spectrum of functions: the most intensive use of common functions is during the protection, response, and recovery operations. Capabilities for common functions are established during the preparatory work. Every common function reinforces the other functions of the cluster. A similar approach is applied by the ResiStand project, identifying the 'supporting functions' [67]. The Universal Task List issued by the US Department of Homeland Security defines as 'common tasks' preparedness, resource management, communications and information management, and supporting technology [81].

### 3.3.1. Crisis Communications and Information Management

The structuring of the crisis communications and information management (CCIM) functional area is developed within two main assumptions: an integrated communications system is established to provide opportunities for agencies and levels of command and management to communicate; information flows are managed according to a coordinated architecture and procedures [81].

Communications capabilities are organised in networks with a crisis communications system at the core. Information flows are regulated bottom-up as the data and information flow to provide for situational awareness; and top-down as warning and alerting, a common operational picture, and commands and advice; as well as between governmental agencies and volunteer organisations that provide responders, resources, or services. Interoperable communications are seen as a critical resource and capability for crisis management [40].

The communications capabilities also include the social networks used for the dissemination of warning messages, advice, and instructions. Tasks for the integration of the national crisis communications system with the Common Emergency Communication and Information System, managed by the European Commission [44], are also envisioned.

Information is managed to support a complex goal: to provide the decision-makers with substantial and timely information, to store the information in a reliable, secure, and convenient manner, and to

ensure that mission-critical information and situational awareness are effectively distributed between interagency partners and through the crisis management hierarchy [109], (pp. 29–30). Crisis information is provided also to people with disabilities and individuals with limited proficiency in the national language [40].

The taxonomy binds the crisis communications and information management functions and tasks with the command, control, and coordination functional area and with all the operations across the crisis management spectrum, including inter-agency flows and exchange with international partners. Despite that their primary goal is to support the response operations, communications and information flows pass across the protection, response, and recovery functional areas in a continuous process with varying intensity.

### 3.3.2. Command, Control, and Coordination (C3)

The command, control, and coordination (C3) functional area of the taxonomy is designed with the aim to reflect the requirements and processes of effective and timely decision-making and implementation coordination and control at every level of jurisdiction (central, regional, and local) and across the professional and voluntary organisations that provide responders, resources, and services. The command and control components' tasks are formulated with the understanding that, in the case of single agency engagement, full C3 is applied. At the same time, in inter-agency operations, leaders are " … convened to co-ordinate the involved agencies' activities and, where appropriate, define strategy and objectives for the multi-agency response as a whole" [79] (Art. 4.2). The structure of the command and control functions and tasks is adapted from the recommendations of ISO 22320 [88] (Art. 4).

Coordination is a particular challenge for both preparatory and operational functions. During preparation, it is mostly about strategy and policy formulation, the establishment of chains of command and management, the allocation of resources, and capability development. As Tom Christensen and co-authors emphasise, "Coordination efforts are particularly important if preparing for multi-level and cross-sectoral crises, and shared planning and training exercises are often crucial too. Both pose instrumental challenges related to potential disagreements between actors, knowledge constraints and coordination problems" [108].

Command, control, and coordination functions might be organised as a single chain of command, e.g., the U.S. National Incident Management System, which includes an Incident Command System defined as " … a flexible, scalable framework for coordinating multi-organisations response" [110]. Another format is usually based on the highest political engagement (of a prime minister, vice prime minister or equivalent), supported by a Council, with one or two ministries having a leading role, and a single agency as a coordinator (e.g., in Romania, Bulgaria, Slovenia).

### 3.3.3. Logistics

Logistics is a common functional area where comprehensive crisis logistics planning and management are organised and conducted, and crisis resources are prepared, stored, and delivered when necessary. Logistics serves primarily to meet the needs of the responders and the affected population. According to the US Federal Emergency Management Agency:

> Logistics integrates whole community logistics incident planning and support for timely and efficient delivery of supplies, equipment, services, and facilities. It also facilitates comprehensive logistics planning, technical assistance, training, education, exercise, incident response, and sustainment that leverage the capability and resources of Federal logistics partners, public and private stakeholders, and nongovernmental organizations (NGOs) in support of both responders and disaster survivors. [111] (p. 1)

The taxonomy defines the functions and tasks for material, transportation, healthcare, and facilities logistics, as well as the provision of core logistic services for the affected people.

### 3.3.4. Security Management

The primary outcome of security management is a secure environment for the responders, people, equipment, and the supplies involved in crisis management operations. Functions and tasks are defined in terms of planning, organising, and capability building for security and safety management. However, the core activities include a provision of public safety and critical infrastructure protection on a day-to-day basis, on-scene security operations in case of an incident, disaster, and crisis, and law enforcement operations and measures to protect affected people, property, and values [40].

## 4. Taxonomy Evolution, Current and Future Use

The first version of the CM taxonomy was created in the fall and winter of 2017, based on the analysis of the academic literature, normative documents, and the experience in relevant research projects, presented in this article. It was also built on the cumulative experience available in the project consortium, including crisis management practitioners, researchers, solution providers, and experts with policy-making experience. The elaboration of the taxonomy benefited from the parallel work on identifying and describing current and future crisis management gaps from the perspective of CM practitioners [112]. In two cycles of scrutiny, consortium partners critically assessed draft versions of the taxonomy and provided numerous recommendations for their improvement. Nevertheless, it was expected that with the start of the practical use of the taxonomy, the accumulation of experience and in-depth knowledge, there will be well founded suggestions for the amendment of the taxonomy—to refine the description of a function, to add a function, or to increase the visibility of a task.

This expectation turned out to be well founded. That taxonomy was incorporated in the online platform "Portfolio of Solutions," supporting the search for solutions to meet identified crisis management needs and the organisation of trials [14]. The feedback from users of the taxonomy, along with new ideas and initiatives, led to a request for the authors to review the taxonomy. The review was conducted in the fall of 2019 and pursued the following objectives:

1. Analyse and account for the experience in the use of the taxonomy in classifying crisis management gaps and solutions on the Portfolio of Solutions (POS) platform;
2. Provide for a classification of new solutions and authoritative lists of crisis management gaps and effective association of gaps and solutions;
3. Expand the taxonomy to allow for a classification of gaps and solutions related to terrorist acts, chemical, biological and radiological threats and the increasing reliance of both society and crisis management organisations on information and communications infrastructures, i.e., on cyberspace.

In addressing the second objective, the team reviewed and searched for the best possible ways to classify the gaps available at the time of the study:

- "Common Global Capability Gaps" identified by the International Forum to Advance First Responder Innovation (IFAFRI), https://www.internationalresponderforum.org/resources; and
- 'strategic gaps' and the challenges, policies, and recommendations elaborated by the EU DRMKC's (Disaster Risk Management Knowledge Centre) "Gaps Explorer" in a pilot project on forest fires, https://drmkc.jrc.ec.europa.eu/knowledge/Gaps-Explorer/forest-Fires.

The review led to version 2.0 of the taxonomy, delivered in November 2019. The high-level structure of the taxonomy—functional areas and functions—was not challenged in the process of revision, and this may be seen as a validation of the approach and the taxonomy's general design. Numerous changes were made at the taxonomy's third and fourth levels—sub-functions and tasks—to provide finer grade classification and the tagging of crisis management gaps and solutions.

At the beginning of 2020, the taxonomy, as part of the POS platform, was translated into Dutch, Estonian, French, German, Italian, Polish, Spanish, and Swedish. The translators posed a number of terminological questions and requests for clarification. As a result, in the current version of the

taxonomy (ver. 2.1), included as a supplement to this article, we reconsidered the use of some terms and attempted to provide descriptions that were clearer to a broader community of stakeholders.

Hence, the current version of the taxonomy is more mature. Although it is designed within a European project, the taxonomy reflects proven international practices codified in the UN, EU, US, and other international guidelines, frameworks, and ISO standards. It is compatible with established international classification schemes (with the exception of specific issues like development aid and proactive functions aimed at preventing terrorism or cyber threats [113,114], see Figure 6). This makes the taxonomy applicable beyond the European context for a variety of purposes.

The CM taxonomy is already well established as the primary tool to facilitate the navigation of the POS platform, match needs and solutions, relate crisis management gaps, trials, solutions, methodological tools, and other kinds of information of interest. However, this is only a fraction of the power such comprehensive taxonomy provides. It has already been in use to facilitate the structured assessment of the societal impact of crisis management solutions [115,116]. The CM taxonomy is seen as a suitable framework to guide the comprehensive examination of disaster risk management strategies and measures [117]. Work is in progress to use the taxonomy as a framework to manage the investments in crisis management and disaster risk reduction. Other potential applications are listed in the introductory section of this article.

These early examples demonstrate the theoretical and practical value of the taxonomy. With its comprehensive coverage of disaster risk mitigation measures, adaptation to climate change and other slow-onset hazards, resilience capacity, and capabilities for protection, response, and recovery, all set in a sound conceptual and decision-making framework, the taxonomy can serve as a platform for further research, elaboration and evaluation of crisis management policy options.

**Supplementary Materials:** The current version of the crisis management taxonomy of functions (ver. 2.1, May 2020) is available online at http://www.mdpi.com/2071-1050/12/12/5147/s1.

**Author Contributions:** The authors have jointly discussed, hotly debated, and finally agreed on the methodology for designing the taxonomy of crisis management functions, the underlying conceptual framework, the overall structure, and the version of the taxonomy presented here. All authors have read and agreed to the published version of the manuscript.

**Funding:** The research leading to these results was performed by the Centre for Security and Defence Management, Institute of ICT, Bulgarian Academy of Sciences, as part of the DRIVER+ project and has received funding from the European Union's Seventh Framework Programme under grant agreement no. 607798.

**Acknowledgments:** The authors gratefully acknowledge the contribution of many researchers in the DRIVER+ project who participated in discussions of early drafts and the review of the taxonomy, and in particular that of Laurent Dubost, Thales Communications & Security S.A, France; Dirk Stolk, TNO, The Netherlands; Martha Bird, Danish Red Cross; Palacio Camilo, Austrian Red Cross; Tomasz Zwęgliński, SGSP, Poland; and most of all Denis Havlik, Austrian Institute of Technology, who relentlessly critiqued the taxonomy, leading at the same time its incorporation in the DRIVER+ online platform "Portfolio of Solutions" and promoting its use by a wide community of stakeholders.

**Conflicts of Interest:** The authors declare no conflict of interest. The funders had no role in the design of the study; in the collection, analyses, or interpretation of data; in the writing of the manuscript, or in the decision to publish the results.

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
