# Peer review of "A Taxonomy of Crisis Management Functions"

_sustainability, doi:10.3390/su12125147_

Round 1
Reviewer 1 Report
Comments and Suggestions for Authors
1.The abstract is needs revision to reflect the core findings of the research.
2.The introduction of the manuscript should be revised to address the research in the field of the crisis management method.
3.This paper needs more clear explain validation and verify of the proposed method.
4. The authors do not provide a convincing significance, What’s the difference of your taxonomy with others’?
Author Response
1.The abstract is needs revision to reflect the core findings of the research.
The main result of our study is the taxonomy itself, and it is most succinctly presented in the abstract (listing its highest level with the ten functional areas). We explained the immediate purpose,a s well as some broader applications already underway. One sentence was added stating that the taxonomy was “designed on the basis of a conceptual model integrating the concepts of hazard, vulnerability, risk, and community, and main consequence- and management-based concepts.” This is the maximum information we were able to provide, given the limitation of 200 words for the abstract.
2.The introduction of the manuscript should be revised to address the research in the field of the crisis management method.
The article is focused on the taxonomy of crisis management functions. We take the functions as defined in national and international norms, standards and guidelines and on that basis structure the taxonomy and describe the taxonomic terms. Applicable crisis management concepts are presented in sub-section 2.2.4 in nearly two pages.
3.This paper needs more clear explain validation and verify of the proposed method.
We are not aware of an applicable theory of designing a taxonomy of this type. We based our design on a clear, in our view, and sufficiently comprehensive conceptual framework, presented in detail. Nevertheless, there are no guarantees that our results are reproducible. Hence, we can claim that it has been validated in the strict scientific sense of term.
The taxonomy has been designed to meet a practical purpose, and it has been scrutinised and used by a broad group of stakeholders for more than 18 months, who provided feedback and recommendations. We added the following statement (p. 29, top): The high-level structure of the taxonomy—functional areas and functions—was not challenged in the process of revision, and this may be interpreted as validation of the approach and the taxonomy’s general design.
- The authors do not provide a convincing significance, What’s the difference of your taxonomy with others’?
In addition to the application examples, presented in the first version of the article, we added a paragraph at the end of the article highlighting the broader theoretical and practical value of the taxonomy.
Existing taxonomies are either very simple (e.g. for the type of hazards, intensity, or geographical coverage) or cover a specific issue of interest, e.g. community interaction [Auferbauer et al., 2019], earthquake disaster response and recovery [Yang and Wu, 2019], communication deployability in disaster management [Pandey and De, 2018]. On the other hand, the taxonomy presented here is:
- comprehensive, covering disaster risk mitigation measures, adaptation to climate change and other slow-onset hazards, resilience and capabilities for protection, response and recovery; and
- functionally oriented, thus providing for straightforward application of state-of-the-art capability planning methods.
Reviewer 2 Report
- As the Special Issue concerns recent developments in the field of risk and crisis management towards realization of the sustainable development goals, Authors should consider to connect the findings with UN sustainable development goals or any other issue of sustainability. It is worth to be mentioned, at least, in the instruction and the discussion chapters.
- There is no information about methodology in the abstract. Such kind of information is crucial to sustain the results and make the abstract structure complete.
- How do Authors interpret the sentence “sustainable use of the taxonomy” (page 2)? What does “sustainable” mean in this context?
- Authors mentioned about “the Community” (page 2). Why did they write it using capital letter? Is this a proper name?
- The methodology is not sound and clearly interpreted. I suggest to modify the Introduction by identification of the research gaps, determination of the research objectives (filling the gaps) and matching particular chapters and subchapters to the objectives and gaps. Also in the conclusion chapter, the research results should be described in the light of the objectives and the gaps, paying special attention to relation with UN sustainable development goals or any other issue of sustainability.
- Lines 79-81 say about 3 elements (1. Information collection; 2. Systemic analysis; 3. Classification of system attributes). However, Fig. 1 visualises 8 elements. How are “the 8 elements” related with “the 3 elements”.
- Authors should define a “testbed” as it is strongly related with Driver+ project nomenclature and not clearly understandable for all readers.
- What are the relations between Fig. 1 with 2.1., 2.2. and 2.3.? I see the strong necessity to define all terms and issues mentioned in 2.1. and 2.2., for sure. However, reader expects sound relation between Driver+ Taxonomy of crisis management functions development approach (Fig. 1) with chapters and subchapters. Without it the content is structured chaotically.
- Text in lines 154-165 is highlighted. What is the reason? Is this a citation? What about a relevant source?
- Authors mentioned that “(…) to differentiate the management at ‘3+’ levels” (page 9). I suggest to enumerate the level names just after this sentence to reduce a potential confusion in interpretation of an entire spectrum of CM terminology (eg. incident management, disaster or emergency management, crisis management).
- “Applying such logic, the mission (crisis management) is realised” (page 13) – in my opinion CM is not a mission but a manner to achieve the mission (understood as a general objective). Thus, I suggest to modify this sentence.
- Text in lines 709-711 is highlighted. I guess this is a citation from [39]. The source should be noted again.
- “Decision-making and operations are the essence of crisis management”. I partially agree. In my opinion (not only mine), crisis management is a kind of management, so decision-making states its essence, for sure. But in case of the operations, they have generally an execution character. There is no management “inside” (commanding -yes, but management – no). Exemplifying, fire service is not a crisis management entity as it operates in time of crisis situation or crisis and executes what crisis manager (public administration body) decide. Thus, I suggest to modify this sentence by delete “and operations are”.
- Can you describe shortly general conclusions which allowed you to review the taxonomy (lines 1093-1102)?
- When discussing CM taxonomy benefits, Authors present the benefits for Driver+ project, mainly. I recommend to modify the methodology to emphasize a new core – not Driver+ itself but gaps and objectives more directly corresponding with UN sustainable development goals or any other issue of sustainability. I am sure, CM taxonomy is (not only can be but truly is) very helpful for CM theoreticians and practitioners. But in this Special Issue, the sustainability issue should be more emphasized. Especially, when CM taxonomy has a great potential to significantly enrich the sustainability theory.
Author Response
Responses to the questions, comments, and recommendations by Reviewer 2

Round 2
Reviewer 2 Report
Authors have answered all my questions. The answers acceptably meet relevant expectations. The methodology soundness has been increased and all necessary corrections have been made.